Blood markers of endothelial dysfunction and their correlation to cerebrovascular reactivity in patients with chronic hepatitis C infection

Pavicic Ivelja Mirela mpavivelj@kbsplit.hr 1
Dolic Kresimir 2
Tandara Leida 3
Perkovic Nikola 4
Mestrovic Antonio 4
Ivic Ivo 1
1 University of Split School of Medicine, University Hospital of Split, Department of Infectious Diseases , Split , Croatia , Croatia
2 University of Split School of Medicine, University Hospital of Split, Department of Radiology , Split , Croatia , Croatia
3 University of Split School of Medicine, University Hospital of Split, Department of Medical Laboratory Diagnostics , Split , Croatia , Croatia
4 University of Split School of Medicine, University Hospital of Split, Department of Gastroenterology , Split , Croatia , Croatia
Davis Joshua
Electronic publication date: 2021 Jan 14
Publication date: 2021
Volume: 9
Electronic Location ID: e10723
Received 2020 Sep 23; Accepted 2020 Dec 16
Copyright: ©2021 Pavicic Ivelja et al.
Copyright year: 2021
Copyright holder: Pavicic Ivelja et al.
License: This is an open access article distributed under the terms of the Creative Commons Attribution License, which permits unrestricted use, distribution, reproduction and adaptation in any medium and for any purpose provided that it is properly attributed. For attribution, the original author(s), title, publication source (PeerJ) and either DOI or URL of the article must be cited.
License URL: https://creativecommons.org/licenses/by/4.0/

Keywords: Hepatitis, Cerebrovascular disorders, Inflammation, Diagnostic imaging, Biomarkers, Atherosclerosis

Funding: The authors received no funding for this work.

==============================
Although liver cirrhosis and hepatocellular carcinoma are major consequences of hepatitis C (HCV), there has been an increasing number of studies examining extrahepatic manifestations, especially those caused by systemic chronic inflammation and metabolic complications that might predispose HCV patients to atherosclerosis and ischemic cerebrovascular disease (CVD). The aim of our study was to assess E-selectin, VCAM-1, ICAM-1 and VEGF-A serum levels in patients with chronic HCV infection and to correlate them with cerebrovascular reactivity. A blood sample was taken from eighteen patients with chronic hepatitis C infection and from the same number of healthy blood donors in the control group. The aim was to analyse markers of endothelial dysfunction and to correlate them with cerebrovascular reactivity expressed as breath-holding index (BHI) determined using transcranial color Doppler. The obtained results revealed significant differences between the groups in all endothelial markers except for the E selectin. While the ICAM-1 and sVCAM-1 were significantly increased in the hepatitis group, VEGF-A was significantly decreased. A significant reduction of 0.5 (95% CI 0.2, 0.8) in the mean BHI was found in the hepatitis group (mean BHI 0.64) compared to controls (mean BHI 1.10). No significant association between the BHI and any of the endothelial markers was found in the control group, while in the hepatitis group, the scatter plot of ICAM-1 vs BHI suggested that the association might be present. In conclusion, the results of this study confirm an association between a chronic HCV infection and altered cerebrovascular reactivity as well as higher levels of markers of endothelial activation (ICAM-1, VCAM-1) as possible indicators of an increased CVD risk.

Introduction

Despite the availability of well-tolerated and effective treatments with novel oral antivirals (direct-acting antivirals, DAA) over the last few years, hepatitis C (HCV) still remains a major global health problem. It is estimated that 71 million people worldwide are chronically infected with HCV, 14 million of which in the European Region (Steffen et al., 2020; Petruzziello et al., 2016). The major burden of the chronic HCV infection comes from liver cirrhosis, hepatocellular carcinoma and liver transplantation. However, extrahepatic manifestations of chronic hepatitis C are increasingly being studied. Some of them are immune mediated and others are driven by systemic chronic inflammation and metabolic complications that might predispose patients to atherosclerosis, including cerebrovascular atherosclerosis (Cacoub et al., 2016). Cerebrovascular disease (CVD) is a major health challenge with an increasing number of CVD-related deaths and disabilities. Conventional risk factors for CVD include ageing, smoking, alcohol consumption, obesity, hyperlipidemia, hypertension, atrial fibrillation and diabetes. In addition to the aforementioned, new risk factors such as infectious agents and inflammation, have been documented (Feigin et al., 2014; Hu et al., 2017; Boehme, Esenwa & Elkind, 2017). Although numerous studies have linked HCV to a higher prevalence of CVD, this relationship, as well as the assumed pathophysiological mechanism, still remain controversial. HCV is generally thought to stimulate pro-inflammatory cytokine synthesis leading to pro-atherogenic activity (Lee et al., 2010; Liao et al., 2012; Huang, Kang & Zhao, 2014; Ambrosino et al., 2016). Consequently, E-selectin, intracellular adhesion molecule-1 (ICAM-1), vascular cellular adhesion molecule-1 (VCAM-1) as markers of endothelial activation, as well as vascular endothelial growth factor -A (VEGF-A) as the regulator of angiogenesis, have been found to be potential indicators of endothelial dysfunction, atherosclerosis and risk of CVD (Stanimirovic et al., 1997; Blann et al., 1999; Poggesi et al., 2016; Billinger et al., 2017; Tchalla et al., 2017; Setyopranoto et al., 2019; Varona et al., 2019).

Ultrasound data were collected as previously described in Pavicic Ivelja et al. (2019) using transcranial color Doppler sonography (TCCD) which allows a rapid and non-invasive investigation of cerebral blood flow (CBF) velocities in the large cerebral arteries and velocity changes after a vasodilatory stimulus such as apnea. These changes, known as cerebrovascular reactivity, are considered an indicator of the capability of adaptive changes in CBF and can be reduced in patients under risk of CVD (Markus & Harrison, 1992; Silvestrini et al., 1996; Apruzzese et al., 2001).

Since in previous studies HCV was found to be associated with higher prevalence of stroke, the aim of our study was to assess E-selectin, VCAM-1, ICAM-1, VEGF-A serum levels in patients with chronic HCV infection and to correlate them with cerebrovascular reactivity assessed by breath-holding technique.

Material and Methods

An cross-sectional observational study was performed at the University Hospital of Split, Croatia after being granted by the University Hospital of Split Ethics Committee (Reg. No.: 2181-147-01/06/J.B.-14-2). Written informed consent was obtained from all participants. Two groups of participants were investigated: group one included 18 chronic hepatitis C patients and group two consisted of 18 healthy blood donor volunteers whose blood is regulary tested for blood-borne diseases including HCV. Patient’s medical history included past and current diseases, laboratory tests, including blood sugar and lipid profile done within three months prior study entrance, and information on alcohol consumption and smoking. Exclusion criteria were: history of hypertension, diabetes mellitus, CVD, hematologic disease, cirrhosis, chronic heart disease, malignancy, excessive alcohol consumption defined as more than 7 drinks per week for women and more than 14 drinks per week for men, as well as therapy with β-blocking agents, hormonal substances, nitrates, calcium channel blockers, vasodilatory drugs and anticoagulants (Rimm et al., 1996; Mukamal et al., 2010; Stoutenberg et al., 2013). Because of inadequate insonation of their middle cerebral artery (MCA), two participants, one from each group, were excluded from the study. The groups were balanced by age, BMI, sex, and alcohol intake (Table 1). In the hepatitis group the estimated mean ± SD duration of disease was 14.6 ± 5.5 years. However, the only significant difference between the control and the hepatitis group was the prevalence of smoking with the odds of smoking in hepatitis group being ∼9 higher than in controls (OR 9.1, 95% CI [1.64–55.10]) Table 1.

Table 1 Sociodemographic and behavioral characteristics of participants by group.

	Control (N = 18)	Hepatitis (N = 18)	P-value	
Age, mean ± SD	35.3 ± 6.5	38.4 ± 4.8	0.111a	
BMI, median (IQR)	25.9 (4.8)	25.8 (5.3)	0.226b	
Sex, N (%)				
Sex 0e	1 (6%)	3 (17%)	0.603c	
Sex 1	17 (94%)	15 (83%)		
Smoking, N (%)				
No	13 (72%)	4 (22%)	0.007c	
Yes	5 (28%)	14 (78%)		
Alcohol				
No	10 (56%)	15 (83%)	0.146c	
Yesd	8 (44%)	3 (17%)		
Notes.

a Independent samples t-test

b independent Mann-Whitney U test

c Fisher exact test

d per week more than seven drinks for women and 14 drinks for men

e sex 0-female, 1-male

Venous blood samples were collected from all participants included in the study. Blood was collected in serum tubes and then centrifuged. Serum aliquots for determination of E-selectin, ICAM-1, VCAM-1 and VEGF-A were stored at −20 °C until analysis. Biochemical analyses were performed using the enzyme-linked immunosorbent ELISA assay (eBioscience, Vienna, Austria).

All subjects underwent TCCD assessment using the Acuson X500 Ultrasound system (Siemens, Erlangen, Germany) with a P4-2 (2-4 MHz frequency) transducer during afternoon in a quiet room while lying in a comfortable supine position after 5 minutes of bedrest. After measuring their blood pressure, the dominant side circle of Willis arteries was insonated, which means exposed to ultrasound waves, through the temporal bone window with the special focus on the MCA. Cerebrovascular reactivity to hypercapnia was evaluated by means of well-established breath-holding test with MCA blood flow mean velocity (Vmean) first measured at rest and then continuously during the test (Markus & Harrison, 1992; Silvestrini et al., 1996; Settakis et al., 2002; Rodriguez-Flores et al., 2014). Participants were asked to hold their breath following normal inspiration for as long as they could. MCV Vmean and duration of breath-hold measured in seconds were recorded. The whole procedure was repeated after 3 to 4 minutes of rest. Mean values of each variable from both measurements were determined and then taken into calculations for breath-holding index (BHI), a ratio between the percentage of increase of Vmean during breath-holding and the duration of breath-holding.

Statistical analysis

Statistical analysis was performed using the SPSS package (version 24.0; Chicago, Illinois). Fisher exact test was used to test the difference in distribution of qualitative data between the hepatitis and control group. For quantitative data, depending on the data distribution, either Mann-Whitney U tests or t-test for independent samples were used. Multiple linear regression models of a endothelial marker level with both hepatitis group and smoking status as independent variables were used to corroborate if the observed difference in a endothelial marker level between the groups is independent of a smoking status. To investigate correlation between the BHI and endothelial markers we calculated Pearson’s correlation coefficients. The significance level was set at 0.05.

Results

Markers of endothelial dysfunction as well as BHI in hepatitis C patients and controls are presented in Table 2. There were significant differences between the groups in all the endothelial markers except for E selectin. While the ICAM-1 and sVCAM-1 significantly increased in the hepatitis group, VEGF-A significantly decreased. Table 2 Observed differences in endothelial marker level remained significant even after adjusting for smoking (regression models; P < 0.020 for ICAM-1, sVCAM-1 and VEGF-A) Table 3.

Table 2 Endothelial markers and BHI distributions by study group.

	Control	Hepatitis	P-value	Mean/median difference	
VEGF-A [pg/ml]. mean ± SD	928.07 ± 390.03	619.71 ± 260.33	0.009*	308.4 (95% CI 83.7, 533)	
ICAM-1 [ng/ml]. median (IQR)	351.10 (132.86)	535.40 (221.49)	<0.001**	−167.9 (95% CI −261.7, −89.3)	
sVCAM-1 [ng/ml]. median (IQR)	804.66 (245.42)	1026.73 (647.25)	0.002**	−257.6 (95% CI −497.2, −116.4)	
E-selectin [ng/ml]. mean ± SD	69.11 ± 23.30	85.36 ± 39.56	0.145*	−16.3 (95% CI −38.4, 5.9)	
BHI, mean ± SD	1,10 ± 0,39	0,64 ± 0,44	0.005*	0.5 (95% CI 0.2, 0.8)	
Notes.

* Independent samples t-test

** independent Mann-Whitney U test

Table 3 Endothelial marker level differences adjusted for smoking.

	Unstandardized Coefficients B	95,0% Confidence Interval for B	P-value	
		Lower Bound	Upper Bound		
VEGF-A (pg/ml)				
Smoking	19.102	−244.920	283.125	0.884	
Hepatitis C	−317.913	−581.528	−54.298	0.020	
ICAM-1 (ng/ml)				
Smoking	25.215	−65.681	116.111	0.576	
Hepatitis C	169.386	78.631	260.142	0.001	
sVCAM-1 (ng/ml)				
Smoking	−480.984	−1.086.514	124.546	0.116	
Hepatitis C	844.065	239.471	1.448.660	0.008	
E-selectin (ng/ml)					
Smoking	8.202	−17.493	33.897	0.521	
Hepatitis C	12.151	−13.505	37.807	0.342	
BHI					
Smoking	.085	−.272	.442	.629	
Hepatitis C	−.502	−.859	−.145	.008	

A significant reduction of 0.5 (95% CI 0.2, 0.8) in the mean BHI was found in the hepatitis group (mean BHI 0.64) compared to controls (mean BHI 1.10) (Table 2). No significant association of BHI with any of endothelial markers was identified in the both groups studied Table 4.

Table 4 Association of BHI with endothelial marker levels by study group.

			VEGF-A (pg/ml)	ICAM-1 (ng/ml)	sVCAM-1 (ng/ml)	E-selectin (ng/ml)	
Control	Pearson Correlation	−0.043	0.005	−0.054	0.070	
	95% Confidence Interval	Lower	−0.506	−0.583	−0.451	−0.539	
		Upper	0.488	0.619	0.391	0.567	
	P-value		0.884	0.986	0.856	0.813	
	N		14	14	14	14	
Hepatitis	Pearson Correlation	0.085	0.430	0.103	0.284	
	95% Confidence Interval	Lower	−0.254	−0.262	−0.194	−0.342	
		Upper	0.429	0.799	0.619	0.705	
	Sig. (2-tailed)		0.737	0.075	0.684	0.254	
	N		18	18	18	18	

However in the hepatitis group the scatter plot of ICAM-1 vs BHI suggests that the association might be present (Fig. 1). Nevertheless, due to the small sample size, the pattern on the scatter plot is not clear and the association might reflect a false positive finding Fig. 1.

Figure 1 Scatter plot of ICAM-1 vs BHI by control group.

Discussion

Our study showed that cerebrovascular reactivity values, calculated as the BHI, were significantly lower in the HCV group than in the healthy control group. Similar results were obtained in our previous study which consisted only of ultrasound parameters, emphasising reduction of vasodilatatory capacity of the cerebral arteries in patients with chronic hepatitis C (Pavicic Ivelja et al., 2019). The above mentioned findings suggest an unfavorable effect of chronic HCV infection on cerebrovascular hemodynamics and might lead to an increased risk of CVD. However, the pathogenesis of the assumed HCV and CVD connection, although probably involving atherosclerosis, is still mostly undefined (Rosenfeld & Campbell, 2011; Masiá et al., 2011). Chronic infections are assumed to stimulate a prolonged systemic inflammatory response leading to blood vessel endothelium damaging and eventually to atherosclerosis development (Campbell & Rosenfeld, 2015). Therefore, markers of endothelial activation (E-selectin, ICAM-1, VCAM-1) and the regulator of angiogenesis VEGF-A were included in our research. The obtained results show that hepatitis C group had significantly higher concentrations of ICAM-1 and sVCAM-1 compared to controls which is in accordance with most results of previous studies dealing with HCV, endothelial dysfunction and higher risk of CVD (Stanimirovic et al., 1997; Blann et al., 1999; Martinic-Popovic et al., 2014). Authors of some recent studies went a step further alleging that the new DAA treatment reverses these chronic HCV effects and enhances endothelial function (Schmidt et al., 2018; Davis et al., 2018; Muñoz Hernández et al., 2020). In our research, HCV patients had decreased VEGF-A compared to healthy controls opposing several earlier studies connecting higher levels of VEGF-A with the risk of ischemic CVD (Shoamanesh et al., 2016). Salum et al. in their study showed that patients with chronic HCV infection and late fibrotic stages of liver disease had the highest levels of VEGF-A compared to healthy controls (Salum et al., 2017). Since the advanced stage of liver fibrosis, regardless of its etiology, is characterized by hemodynamic alterations, cirrhotic patients were not included in our study. This is likely the reason why we had contrasting results connected with VEGF-A but it could also imply that the concentration of this biomarker may not be relevant if HCV is taken as an independent risk factor for CVD and that it has more to do with the progression of liver fibrosis. Same interpretation could be given for our E-selectin results (no significant differences between the groups, Table 2) since previous research also suggested its association mostly with severity of liver fibrosis (Kaplanski et al., 1997; Montalto et al., 2001). Although we got significant independent results for ultrasound parameters of cerebrovascular reactivity as well as blood markers of endothelial dysfunction, we wanted to combine these two methods as this has not been previously done in the context of chronic hepatitis C and risk of CVD. No significant association of BHI with any of endothelial markers was identified in the control group. However, in the hepatitis group, BHI was moderately associated with ICAM-1 at the 0.1 significance level (Table 4) indicating the need for further studies with a larger number of participants including post-DAA HCV patients. Also, the fact that cirrhotic patients were not included, and still such significant differences in cerebrovascular reactivity and markers of endothelial dysfunction are present, implies the need for active diagnosis and early treatment of HCV patients, especially nowadays with available effective DAA treatment. Thus, not only the progression of liver damage but also possible systemic atherosclerosis induced by chronic inflammation could be prevented. The focus of our work was on cerebral vasoreactivity which probably reflects systemic impairment of the vascular system rather than isolated cerebrovascular dysfunction. In that sense it would be interesting in future research to evaluate established methods in assessing peripheral vascular endothelial function like brachial artery flow-mediated dilation (FMD) and reactive hyperemia-peripheral arterial tonometry (RH-PAT) in HCV patients (Staszewski et al., 2019).

The main limitation of the study is a small sample size which allowed only for inclusion of limited number of predictors in multiple linear regression (HCV status and smoking Yes/No), and thus hindered investigation of possible effects of confounding variables. A much larger sample size and multiple regression model with more factors would be required in future studies to mitigate this effect. Although we excluded examinees with clinical, laboratory and ultrasound features of cirrhosis from the study we did not preform fibrosis score which could also be considered as a study limitation.

In conclusion, BHI findings in our study suggest that patients with chronic hepatitis C have an altered cerebrovascular reactivity and a higher risk of unfavourable cerebrovascular events. Additional evidence has also been provided to link the above-mentioned markers of endothelial dysfunction and CVD risk in patients with chronic HCV infection. The question remains whether the level of these markers, or at least one of them, combined with BHI could potentially be used in assessing the risk of CVD in patients with chronic hepatitis C. If proven to be useful after additional studies, endothelial markers together with BHI could be of great assistance in the comprehensive evaluation of HCV patients and CVD prevention.

Supplemental Information

Data S1 Raw data

Click here for additional data file.

Supplemental Information 2 Codebook

Click here for additional data file.

Additional Information and Declarations

Competing Interests

Author Contributions

Human Ethics

Data Availability

The authors declare there are no competing interests.

Mirela Pavicic Ivelja conceived and designed the experiments, performed the experiments, analyzed the data, prepared figures and/or tables, authored or reviewed drafts of the paper, and approved the final draft.

Kresimir Dolic conceived and designed the experiments, authored or reviewed drafts of the paper, and approved the final draft.

Leida Tandara performed the experiments, analyzed the data, prepared figures and/or tables, authored or reviewed drafts of the paper, and approved the final draft.

Nikola Perkovic and Antonio Mestrovic analyzed the data, prepared figures and/or tables, and approved the final draft.

Ivo Ivic conceived and designed the experiments, analyzed the data, authored or reviewed drafts of the paper, and approved the final draft.

The following information was supplied relating to ethical approvals (i.e., approving body and any reference numbers):

The University Hospital of Split, Croatia Ethics Committee granted ethical approval (Reg. No.: 2181-147-01/06/J.B.-14-2).

The following information was supplied regarding data availability:

The raw measurements and codebook are available in the Supplemental Files.

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
