# Peer review of "Blood markers of endothelial dysfunction and their correlation to cerebrovascular reactivity in patients with chronic hepatitis C infection"

_PeerJ, doi:10.7717/peerj.10723_

## Round 0.1 · original submission · Minor Revisions

1. Remove the sentence "the BHI was moderately associated with the ICAM-1 at the 0.1 significance level". If the p value was 0.1, then there was no significant association, and this p-value could easily occur by chance with so many comparisons.

2. Please explain the term "insonated" in the text at first use

3. Stats analysis - Fisher's exact test cannot be used to compare qualitative data - please re-word this. (I think you are using "qualitiative" to mean categorical variables and "quantitative" to mean continuous?)

4. Please expand the discussion as per the reviewers' suggestions and, in addition:
4.1 Please describe and cite other methods of testing vascular reactivity (e.g. FMD, RH-PAT), and discuss what cerebral BHI assessment adds to these - how is it different, does it correlate etc?

4.2 Please cite other studies that have measured blood markers of endothelial activation in HCV e.g.

Muñoz-Hernández R, Ampuero J, Millán R, Gil-Gómez A, Rojas Á, Macher HC, Gallego-Durán R, Gato S, Montero-Vallejo R, Rico MC, Maya-Miles D, Sánchez-Torrijos Y, Soria IC, Stiefel P, Romero-Gómez M. Hepatitis C Virus Clearance by Direct-Acting Antivirals Agents Improves Endothelial Dysfunction and Subclinical Atherosclerosis: HEPCAR Study. Clin Transl Gastroenterol. 2020 Aug;11(8):e00203. doi: 10.14309/ctg.0000000000000203. PMID: 32955194; PMCID: PMC7431267.

Schmidt FP, Zimmermann T, Wenz T, Schnorbus B, Ostad MA, Feist C, Grambihler A, Schattenberg JM, Sprinzl MF, Münzel T, Galle PR. Interferon- and ribavirin-free therapy with new direct acting antivirals (DAA) for chronic hepatitis C improves vascular endothelial function. Int J Cardiol. 2018 Nov 15;271:296-300. doi: 10.1016/j.ijcard.2018.04.058. Epub 2018 Aug 1. PMID: 30077529.

Davis JS, Young M, Lennox S, Jones T, Piera K, Pickles R, Oakley S. The effect of curing hepatitis C with direct-acting antiviral treatment on endothelial function. Antivir Ther. 2018;23(8):687-694. doi: 10.3851/IMP3257. PMID: 30048244.

4.3 In limitations, please discuss the fact that the results could be cofounded by unmeasured systematic differences between patients with HCV and those without it. To control for these adequately would require thousands of patients in each group and a multiple regression model with many factors.

Reviewer 1 ·

Basic reporting

Well written paper, easy to follow, well structured and to the point. Adequate references.

Experimental design

Chronic hepatitis C is associated with ischaemic stroke. The mechanism may relate to pro-inflammatory cytokine synthesis leading to pro-atherogenic activity. This paper looks to link increased levels of E-selectin; ICAM-1; VCAM-1; VEGF-A (as markers of inflammation) in HCV patients with reduced cerebral vasodilatory response to hypercarbia (as a marker of progressive atherosclerosis).

Validity of the findings

There are a few points to address:
1) BHI would increase due to progressive carotid stenosis and intracranial atherosclerosis, neither mechanisms yet demonstrated to be more relevant in HCV patients than any other cause of stroke. The relevance of this study findings relies on evidence that atherosclerosis causes most strokes in HCV patients and this may be worth mentioning also in the discussion. Were these patients screened for extracranial carotid stenosis?
2) This study is likely to have identical findings when comparing 18 chronic smokers with 18 non-smokers. Using smoking as an exclusion criterial for this study may result in more convincing findings. Nevertheless, it would be useful to include more detail about the multiple regression.
3) It would be prudent to know how Hepatitis C was acquired, or more relevantly if any or many of the 18 HCV group used injectable methamphetamines (also known for increasing the risk of cerebrovascular events)?

Reviewer 2 ·

Basic reporting

Overall an interesting paper with clinical relevance.

Few small comments:

In table 3, E-selectin is spelt wrongly, and all the commas should be full stops.

The sentence in the abstract and results "A significant difference in the BHI was found in the hepatitis group showing lower values of 0.5 (95% CI 0.2, 0.8) on average" is a bit unclear. It should read: "A significant reduction of 0.5 (95% CI 0.2, 0.8) in the mean BHI was found in the hepatitis group (mean BHI 0.64) compared to controls (mean BHI 1.10)"

Experimental design

in line 76 it is stated that this is an interventional and cross-sectional study. However it is not interventional, I would delete this. No changes to patient treatment or medications are made to deem this 'interventional'.

I think this study might be strengthened by reporting the measured non-invasive markers of fibrosis in patient groups, such as the FIB-4 or APRI score. If, as the author state in line 162-169, liver fibrosis may be correlated with changes in endothelial markers, it would be important to quantify differences in such fibrosis scores between groups (even in the absence of frank F4 cirrhosis). It would be important to include these fibrosis scores in regression analyses to ensure that fibrosis is not mediating the differences in endothelial markers between groups.

If the authors don't have this fibrosis score, this is something that should be explicitly stated in the limitations of the study.

Validity of the findings

The discussion section should include a limitations section.

---

## Round 0.2 · accepted · Accept

All important comments have been responded to and adequately addressed, and the paper is substantially improved.